# New Genus and Species of Limnichines from the Cretaceous Amber of Myanmar and Taxonomic Notes on the Family Limnichidae (Coleoptera, Polyphaga) [note 1]

**DOI:** 10.3390/insects13100891

**Published:** 2022-09-30

**Authors:** Alexander G. Kirejtshuk, Alexander A. Prokin

**Affiliations:** 1Zoological Institute, Russian Academy of Sciences, Universitetskaya emb., 1, 199034 St. Petersburg, Russia; 2Papanin Institute for Biology of Inland Waters Russian Academy of Sciences, 152742 Borok, Russia

**Keywords:** Limnichinae, Limnichini, Dryopoidea, fossil records, amber inclusions

## Abstract

**Simple Summary:**

Here is described a new genus and species of minute marsh-loving beetles (Limnichidae) from the Cretaceous Burmese amber. The new fossil can be accurately assigned to the subfamily Limnichinae and tribe Limnichini. The new genus is similar to extant members of the genus *Platypelochares*, currently distributed in the Indo-Malayan Region, and a member of the fossil genus *Hernandochares*
**gen. nov.** from the Eocene Baltic amber. The Mesozoic and Cenozoic fossil genera described here and some other modern limnichines (Polyphaga) have small body size and some structural similarities to myxophagan torridincollids (Myxophaga), probably because of their association with riparian and hygropetric habitats.

**Abstract:**

*Burmochares groehni***gen.** et **sp. nov.,** the oldest known representative of the subfamily Limnichinae and tribe Limnichini, is described from the Cretaceous amber of northern Myanmar. The new genus *Hernandochares*
**gen. nov.** is proposed for *Platypelochares electricus* Hernando, Szawaryn et Ribera, 2018 (type species of this new genus) from the Eocene Baltic amber. The structurally similar genera *Platypelochares,*
*Burmochares*
**gen.**
**nov.**, *Hernandochares*
**gen. nov.**, some other limnichines (Polyphaga) and some torridincollids (Myxophaga) are thought to be connected by their preference for similar habitats and lifestyle.

## 1. Introduction

The family Limnichidae represents a comparatively small but rather diverse group in the Recent fauna with more than 380 species arranged in 37 genera [1,2] from four subfamilies, established by Hinton [3]. In addition, *Palaeoersachus bicarinatus* Pütz, Hernando et Ribera, 2004 and *Platypelochares electricus* Hernando, Szawaryn et Ribera, 2018 were described from the Eocene Baltic amber [4,5]; *Erichia cretacea* Yu, Slipinski, Dong et Pang, 2018 and *Anomocephalobus liuhaoi* Li, Jäch et Cai, 2022 [6,7] were described from the Cretaceous Burmese amber, although limnichids are comparatively frequent among Baltic, Burmese and Dominican [5,8] amber inclusions waiting in different collections for study and description. This paper aims to describe a new limnichid genus and species, *Burmochares groehni*
**gen.** et **sp. nov.** of the subfamily Limnichinae *sensu stricto* from the Cretaceous Burmese amber and a new fossil genus *Hernandochares*
**gen. nov.** (type species *Platypelochares electricus* Hernando, Szawaryn et Ribera, 2018) from the Eocene Baltic amber with small body size and rather peculiar structures, which are similar to those in the Recent genus *Platypelochares*.

## 2. Material and Methods

The paper is devoted to the description of one fossil inclusion in amber, deposited in the Leibniz-Institut zur Analyse des Biodiversitätswandels, Hamburg (further **LIBH**) (before 2021—Geologische-Paläontologisches Institute, Museum, Hamburg; further **GPIMH**), which was collected and pictured by Carsten Gröhn with number (further Collection Carsten Gröhn Glinde—**CCGG**) “CCGG no. 20037”. This amber piece was collected from the amber mines situated in the Hukawng Valley of Kachin State, Myanmar (26°21′33.41″ N—96°43′11.88″ E) (see Geological Setting). The amber piece containing the studied inclusion was manually cut using denticulate shaving blades and then polished with the use of oxide of rare-earth metal cerium. The specimen under description was examined by Carsten Gröhn using a Zeiss stereomicroscope (self-made construction with different objectives: Nikon M Plan 5×, 10×, 20×, 40×; Luminar 18 mm, 25 mm, 40 mm) and an attached Canon EOS 450D digital camera. Line drawings were prepared by the author using Adobe Photoshop CS3.

### Geological Setting

The amber piece with inclusion originated from mines in the Hukawng Valley in the state of Kachin in Myanmar. The fossil resin has been dated stratigraphically and radiometrically from late Albian to early Cenomanian [9]. According to the traditional interpretation [10], the probable Cenomanian radiometric age of Burmese amber has been estimated as taken from sedimentary beds, indicating that it had been re-deposited. Much of the resin is rolled and bored by pholadid bivalves, which demonstrates that the resin was hard before it was buried [11]; the centers of the resin pieces, however, were still soft when bored, so the formation of the amber is considered to be contemporaneous with the deposition of the bed [12,13,14]. Thus, the age of this amber still remains unclear, although an enormous concentration of amber with inclusions in certain geological layers gives evidence that its deposition occurred under peculiar conditions and during a definite term. Mao et al. [15] alleged a primary deposition of this amber in marine circumstances; however, it is still impossible to explain a very considerable accumulation of resin under such conditions without essential addition of fossil resin transported by river water from large territories. Considering these circumstances, the amber material from this locality has to be considered approximately coeval with the amber-bearing deposits, that is at the earliest Cenomanian in age. Nuclear magnetic resonance spectra and the presence of araucaroid wood fibers in amber samples from the Noije Bum 2001 Summit site indicate an araucarian (possibly *Agathis*) tree source for the amber, e.g., [16,17], etc.

## 3. Results

### Systematic Palaeontology

**Order Coleoptera** Linnaeus, 1758

**Suborder Polyphaga** Emery, 1886 [18]

**Infraorder****Elateriformia** Crowson, 1960 [19]

**Family Limnichidae** Erichson, 1845 [20]

**Subfamily Limnichinae** Erichson, 1845 [20]

**Remarks.** The general appearance of the holotype of the new Cretaceous species (oval to streamlined body); features of its head with characteristic appendages (largely exposed labrum and comparatively long subfiliform to only somewhat loosely clubbed antennae); structures of the thoracic sclerites with weak serration along posterior edge on pronotum, short prosternum and expressed interlocking mechanism; moderately widely separated procoxae, extremely widely separated mesocoxae and rather narrowly separated metacoxae with raised characteristic metacoxal plates; fused basal abdominal ventrites and some other structures show its certain relation to some groups of Dryopoidea Billberg, 1820 (1817) [21], but not to Byrrhidae Latreille, 1804 [22] (see also [3] and discussion below), i.e., Limnichidae, Lutrochidae Kasap et Crowson 1975 [23], Dryopidae Billberg, 1820 [21] and Chelonariidae Blanchard, 1845 [24].

*Burmochares***gen. nov.** having a considerable similarity in its general appearance, differs from the lutrochids and dryopids in the subvertical head bearing vertical eyes not projecting laterally and very long subfiliform antennae with very loose club, very short pronotum with deeply trapezoidally excised anterior edge, very short precoxal part of prosternum, widely dilated basal part of the inner edge of epipleura, subcontiguous metacoxae. The chelonariids have a head and prosternum rather similar to those in the new Cretaceous genus, however, the latter is very distinct from the chelonariids in the vertical and not projecting eyes, subfiliform antennae, pronotum with deeply trapezoidally excised anterior edge, prosternum without subparallel paramedian grooves for receipt antennae and widely dilated anterior potion of the inner edge of epipleura.

The new genus should be regarded in the composition of the tribe Limnichini *sensu stricto* of the subfamily Limnichinae because of its characteristic body shape (oval to streamlined), antennal insertions widely separated and covered lateral dilatation of the frons, well-raised grooves for reception of all legs and also tarsal formula 5-5-5 (Figure 1, Figure 2 and Figure 3; see description). However, the described fossil species has no transverse ridge visible at the lateral anterior end of the procoxal cavity [3,25]. However, another subfamily with this tarsal formula is Cephalobyrrhinae Champion, 1925 [26,27], which is also characterized by the absence of the prohypomeral ridge before lateral end of the procoxal cavity, includes members with the more or less elongate body shape with the steeply sloping pronotal and elytral sides, ultimate maxillary palpomere narrowing apically, elongate scutellum, moderately separated mesocoxae and no grooves on ventral surface for leg reception. Another two limnichid subfamilies, Hyphalinae Britton, 1971 [28] and Thaumastodinae Champion, 1924 [29], have the more or less elongate body shape with the steeply sloping pronotal and elytral sides, and also the tarsal formula 4-5-5 or 4-4-4. Additionally, Hyphalinae are characterized by small subovoid procoxae, and Thaumastodinae by oblique metacoxae with rather developed plates. This new genus has no deep concavity on each side of the pronotum for reception of antenna and, therefore, should be certainly attributed to Limnichini *sensu* Spangler, 1999 [30].

**Genus *Burmochares*** Kirejtshuk et Prokin, **gen. nov.**


http://zoobank.org/urn:lsid:zoobank.org:act:1F0B129B-0234-439D-A37E-D641A69D8DB1.

Type species: *Burmochares groehni*
**sp. nov.**

**Etymology.** The name of this new genus is formed from the old name of the country of origin of the holotype of the type species (Burma) plus the end of the generic name “chares”, referring to some similarity of the new genus to some other members of Limnichidae. Gender masculine.

**Diagnosis.** Body nearly evenly oval to streamlined with maximum width in basal third of elytra, moderately convex dorsally and ventrally; upper and lower body integument with rather dense, short and slightly conspicuous hairs; head about two fifths as wide as pronotum at base, smooth from above and with height greater than width (Figure 1a,b); eyes well developed, comparatively small and not projecting laterally (not visible from above), vertical, with above edge subrectilinear and subparallel groove for insertion of antenna (Figure 3a,b); ultimate maxillary palpomere fusiform and ultimate labial palpomere subflattened and widened to truncate apex; antennae rather long, with thick and short scape and pedicel, antennomeres 3–7 subcylindrical to subconical, antennomere 8–11 forming loose club, including antennomeres 8–10 subconical and antennomere 11 subconiform with acute at apex (Figure 3); pronotum gently sloping at sides, with anterior edge trapezoidally excised and anterior angles far projecting, and posterior edge finely and densely serrate, deeply concave and also sinuate at each scutellar side, posterior angles subacute and far projecting posteriorly; subpentagonal scutellum (Figure 1a and Figure 3d); elytra complete, gently sloping to subexplanate sides, apices conjointly rounded (Figure 1a); underside with distinct excavations for insertion of legs; prosternum medially convex, anterior edge very deeply incised and without either serration or clear collar at anterior edge, prosternal process moderately wide and with angular apex; prohypomera with triangular depression at mesoposterior angle; procoxae transverse; metepisterna moderately wide and widening anteriorly; metaventrite with distinct discrimen and without trace of premetacoxal (katepisternal) sutures; metacoxae very narrowly separated; epipleura rather wide at base, apparently strongly arcuately dilated and covering anterior part of metepisternum; abdomen with five ventrites, basal ventrites fused (Figure 1b–d and Figure 2b); legs moderately wide and long, meso- and metatarsi rather long (about four fifths as long as corresponding tibiae); tarsal formula 5-5-5 (Figure 1b and Figure 2c,d).

**Comparison.** The genus *Burmochares*
**gen. nov.** seems to be rather distinct among limnichine members in the shortest prothoracic segment, and, respectively, the rather short precoxal part of prosternum, deeper excision of prosternal anterior edge and longest prosternal process (nearly reaching the level of middle of mesocoxae). This new genus is similar in the body shape, structure of head, eyes, antenna and pronotum, and probably closely related to *Platypelochares* Champion, 1923 [30,31,32] and *Hernandochares*
**gen. nov.** (see below) but differs from both genera in the moderately convex dorsum, much shorter pronotum, deeply excised anterior edge of prosternum, dilatation of epipleura covered part of metepisterna, meso- and metatarsi markedly longer (about four fifths as long as corresponding tibiae) and articulated with tibiae apically (not ventrally laterally); and also from the first in the distinctly lobed and deeply incised labrum, long ultimate maxillary palpomere, subpentagonal scutellum, absence of premetacoxal (katepisternal) sutures; and also from the second genus in the distinctly acuminate posterior angles of pronotum.

**Composition**. The type species only.

***Burmochres groehni*** Kirejtshuk et Prokin, **sp. nov.**
Figure 1, Figure 2 and Figure 3. 

http://zoobank.org/urn:lsid:zoobank.org:act:0C3DB63D-5704-4D93-817C-C6AE939D2606.

**Type material.** Holotype (GPIH no. 5071, CCGG no. 20037, LIBH), a complete beetle, probable female (Figure 2a) is included in an amber piece somewhat viewed as a triangle with sides 18 × 18 × 11 mm and with approximate thickness 2 mm, and also with some facets at edge obliquely smoothed. The body of the beetle could have its head somewhat extracted from the prothoracic segment and genitalia also extracted outside post mortem (Figure 2a). The amber piece has some amber layers around the specimen, producing some small optic aberrations and also many pieces of organic matter of different structures and coloration near the beetle, one subglobular organic piece behind the beetle and of comparable size, and also one longitudinal conglomeration of small pieces of organic matter on the left side from the beetle.

**Type locality and horizon.** North Myanmar, Kachin, Tanai Township, Hukawng Valley; early/late Cretaceous, latest Albian/earliest Cenomanian.

**Etymology.** The name of this new species is dedicated to Carsten Gröhn, great enthusiast in collecting of insect inclusions in amber of different ages, curating the collection of the Leibniz-Institut zur Analyse des Biodiversitätswandels, Hamburg, University of Hamburg (GPIMH).

**Description.** Body length 2.85, width 1.71 mm; moderately and almost evenly convex dorsally and ventrally, subunicolorous dark reddish to brownish. Head with dense regular and shallow punctures markedly lesser than facets in diameter, interspaces between punctures about one puncture diameter and alutaceous (Figure 1c and Figure 3a,c). Pronotum and elytra with distinct punctures slightly sparser than those on head; interspaces between them smoothed (Figure 1a and Figure 3d,e). Sclerites of underside with much coarser and sparser punctures (particularly on metaventrite), interspaces between them more or less smoothed to completely smooth on metaventrite (Figure 1b,d). Upper integument with short, moderately dense and scarcely visible hairs, underside with sparse and less conspicuous hairs.

Head about two fifths as wide as pronotal base and considerably extracted from prothoracic segment, with height greater than width, dorsally gently and rather convex along middle and anteriorly, with very raised border of frons around antennal insertions and dorsally continuing as dilatations over eyes. Eyes exposed mostly laterally (scarcely visible from above), moderately coarsely facetted (nearly twice as coarse as one puncture diameter), with vertical diameter apparently greater than longitudinal diameter. Clypeus distinctly separated by suture from frons. Labrum well exposed from under clypeus and with deep narrow median excision (Figure 1c and Figure 3c). Ultimate maxillary palpomere very long and fusiform. Ultimate labial palpomere subtransverse and subtruncate at apex. Antennae about twice as long as head, with thick and short scape and pedicel (slightly longer than thick), subconical antennomeres 3–10 at least 2.5× as long as thick (antennomeres 3–7 only slightly thickened apically and antennomeres 8–10 more thickened), and antennomere 11 subconiform, somewhat longer than thick, thickest before middle, and with acuminate at apex (left antenna visible from side, but only antennomeres 9–11 visible from above) (Figure 3e).

Pronotum more than twice as wide as long (at sides much longer than along middle), with widely subexplanate sides, rather and evenly convex at disk, arcuate along sides, widest at base, anterior edge trapezoidally excised with lateral sides about as great as transverse eye diameter, posterior edge very finely serrate and posterior angles rather sharp (Figure 1a and Figure 3d). Scutellum slightly wider than long and subpentagonal, with widely rounded apex (Figure 3d). Elytra complete, elongate, about one and a third as long as wide together, conjointly rounded to subangular at apices, sides moderately narrowly explanate; subhorizontal epipleura rather wider at base than metatibiae and gradually narrowing posteriorly, epipleural inner dilatations in anterior two fifths about 1.5× as wide as epipleura at base and covering most of metepisterna and participating in thoracic grooves for reception of mesotibia (Figure 1d and Figure 2d).

Prosternum with very deeply incised anterior edge without visible serration and with moderately long and wide prosternal process, about 2.5× as wide as tibiae. Procoxal cavities rather widely separated and apparently open posteriorly. Mesocoxal cavities oval and large, strongly widely separated, about twice as widely separated as procoxae. Metaventrite with discrimen along whole length. Metacoxae strongly oblique and very narrowly separated to subcontiguous. Metepisterna moderately wide, with posterior end as wide as lateral length of metacoxal plate and rectilinearly becoming wider anteriorly (Figure 1b–d). Abdomen with ventrite 1 longest, about 1.5× as long as each of ventrites 2–4, hypopygidium somewhat longer than each of ventrites 2–4 (ventrite 2 somewhat longer than each of ventrites 3 and 4) but somewhat shorter than ventrite 1 (its apex not clearly visible and apparently widely truncate at apex) (Figure 1b,d and Figure 2b).

Femora and tibiae apparently characteristic in shape of most limnichines with oval body and convex underside. Tibiae narrow and thin, meso- and metatibiae markedly longer than protibia, protibia slightly curved and subparallel-sided, but meso- and metatibiae straight and somewhat gradually narrowing in posterior third, with thin and moderately long spurs. Protarsi slightly shorter and wider than meso- and metatarsi, protarsomeres 1–4 lobed. Meso- and metatarsi comparable in length (about four fifths as long as femora) and proportions of tarsomeres. Four first meso- and metatarsomeres apparently comparable in length and with one comparatively short lobe bearing some setae (clearly visible on metatarsus located somewhat apart from body; other two mesotarsi and another metatarsus approach body and with tarsomeres 1–4 looking simple but with traceable setae on lower surface). Protarsomere 5 about twice, but meso- and metatarsomeres 5 about three times, as long as each of corresponding tarsomeres 1–4; claws rather long (about two fifths as long as tarsomere 5) and somewhat swollen at base (Figure 1b–d and Figure 2c,d).

**Remarks.** The specimen examined has only one antenna completely observable and its basal part is inserted in a groove distinctly separating the upper and lower parts of the eye. The specimen examined demonstrates some structural features that are difficult for a standard morphological interpretation, but worth mentioning for further studies. The prosternum has a narrow stripe along its anterior edge, which is apparently separated from the remaining part of the prosternum by a suture at the sides and interconnected by a furrow in the middle. Perhaps this structure is homologous or homoplastic to the collar in other elateriformians. Additionally, the narrow stripes between the lateral part of the procoxa and lateral part of the anterior isolated stripe along the anterior edge of the prosternum are strongly reminiscent of the propleura in other coleopterous groups, but they are apparently thickened and isolated stripes of the prohypomera. In addition, these stripes between the procoxae and anterior part of prosternum do not reach its anterior edge.

The genitalia behind the abdominal apex are rather membranous, with slightly traced slightly darkened structures and without clear outlines of sclerotization that should be represented in the aedeagus. Thus, it is thought that they appear similar to the “sclerites” of a female rather than male ones (particularly pictured laterally: Figure 2b).

## 4. Discussion

The specimen examined (holotype of *Burmochres groehni*
**sp. nov.**) shows that its epipleura in its anterior part cover the base of the metepisterna, and, therefore, this part cannot be an epipleural rim but presents a dilatation of this rim as a lamella (the weak fold subparallel to the lateral edge of the elytra should be interpreted as a true rim).

The subfamily Dryopoidea includes the following families: Chelonariidae, Dryopidae, Elmidae Curtis, 1840 [33], Protelmidae Jeannel, 1950 [34], Eulichadidae Crowson, 1973 [35], Heteroceridae MacLeay, 1825 [36], Limnichidae, Lutrochidae, Psephenidae Lacordaire, 1854 [37], Ptilodactylidae Laporte, 1836 [38], Callirhipidae Emden, 1924 [39], Cneoglossidae Champion, 1897 [40] and †Mastigocoleidae Tihelka, Jäch, Kundrata et Cai, 2022 [41]. This superfamily was grounded and divided by Crowson into two groups of families (dryopid and elmid groups) and frequently with some differences in composition, argumentation [42,43], etc. Later, different researchers tried many times to find a better interpretation mainly after re-analysis on the basis of the matrix usually adapted from that proposed by Lawrence [44,45], etc. In the above diagnosis of the genus *Burmochares*
**gen. nov.**, only the families including the representatives somehow externally similar in appearance to *Burmochres groehni*
**sp. nov.** were mentioned without consideration of where they were put in different publications.

The single extinct dryopid family, Mastigocoleidae, in contast to both *Burmochares*
**gen. nov.** and *Hernandochares*
**gen. nov.** and other genera considered here, has the elongate body with antennomere 1 the widest and longest, and the remaining antennomeres subcylindrical and gradually thinning apically, an apparently unique antennal structure within Dryopoidea. Mastigocoleids also have the ultimate labial palpomere roughly subcylindrical but obliquely truncate apically and serially punctured elytra.

Additionally, the examined specimen has a set of external characters in the body shape and outlines of its upper and underside sclerites, and also in the leg segments, which are diagnostic for the Deleveinae Endrödy-Younga, 1997 [46] (Myxophaga [47], Torridincollidae Steffan, 1964 [48]). The new fossil limnichid and torridincolids share somewhat similar dilatation of the basal part of epipleura, smoothed dorsal and ventral body integument without sculptural elements, elytra without clear serial punctation, five abdominal ventrites, etc. Nevertheless, the new limnichid genera differ from the torridincollids in the less convex body, narrow and strongly declined subvertical head with markedly vertical and not laterally projecting eyes and rather long antennae, very short prosternum and pronotum, traceable weak serration along posterior edge on pronotum, and also five-segmented tarsi with tarsomeres 1–4 comparable in length and shape. Such a great similarity in appearance of small body and adaptive structures could be caused by the considerable similarity in bionomy and ecology of both groups (modern and probably fossil) associated with riparian and hygropetric habitats.

## 5. Taxonomic Notes

Genus ***Hernandochares* gen. nov**.

Type species *Platypelochares electricus* Hernando. Azawaryn et Ribero, 2018.

http://zoobank.org/urn:lsid:zoobank.org:act:AC7BB9C4-F6AA-43D1-897B-A7CEE20D31BC.

**Etymology**. The name of this new genus is formed from the name of the first author of the type species, C. Hernando, known specialist on limnichids, plus the end of the generic name “chares”, referring to some similarity of the new genus to some other members of Limnichidae. Gender masculine.

**Diagnosis**. Body nearly evenly oval to streamlined with maximum width in basal third of elytra, strongly convex dorsally and ventrally; upper and lower integument with rather dense, semirecumbent and rather conspicuous hairs; smooth from above and with head height greater than width; eyes well developed and apparently vertical; pronotum gently sloping at subexplanate sides, with a regular row of small tubercles in anterior part, extending laterally over the flattened lateral areas, with posterior edge finely and densely serrate, gently convex and without sinuation at scutellar sides, posterior angles widely rounded and not projecting posteriorly; elytra complete, gently sloping at subexplanate sides, apices conjointly rounded, with small apical semicircular excavation, apparently not protruding in middle; underside with distinct excavations for insertion of legs; prosternum medially convex, anterior edge moderately incised, prosternal process moderately wide and with arcuate to angular apex; metepisterna comparatively narrow and curved and somewhat widening anteriorly; metaventrite with feebly expressed discrimen and without trace of premetacoxal (katepisternal) sutures; metacoxae narrowly separated; epipleura rather wide at base, apparently strongly arcuatelly dilated and apparently not covering anterior part of metepisternum; abdomen with five ventrites, basal ventrites fused; legs moderately wide and long; meso- and metatarsi moderately long (less than three fifths as long as corresponding tibiae), articulated with tibiae ventrally to laterally, tarsal formula 5-5-5.

**Comparison**. The genus *Hernandochares***gen. nov.** is similar to *Burmochares*
**gen. nov.** and *Platypelochares* and differs from both in the arcuately excised anterior edge of pronotum, gently convex posterior edge and widely rounded posterior angles of pronotum; and also from the first in the deeply excised anterior edge of prosternum, apex of prosternal process reaching the level of the middle of mesocoxae, epipleural dilatation partly covering anterior part of metepisterna, meso- and metatarsi apically articulated with meso- and metatibiae and about four fifths as long as corresponding tibiae; and also from the first in the meso- and metatarsi articulated with tibiae ventrally to laterally; and also from the second genus in the absence of the premetacoxal (katepisternal) sutures.

**Composition**. The type species only.

## Figures and Tables

**Figure 1 insects-13-00891-f001:**
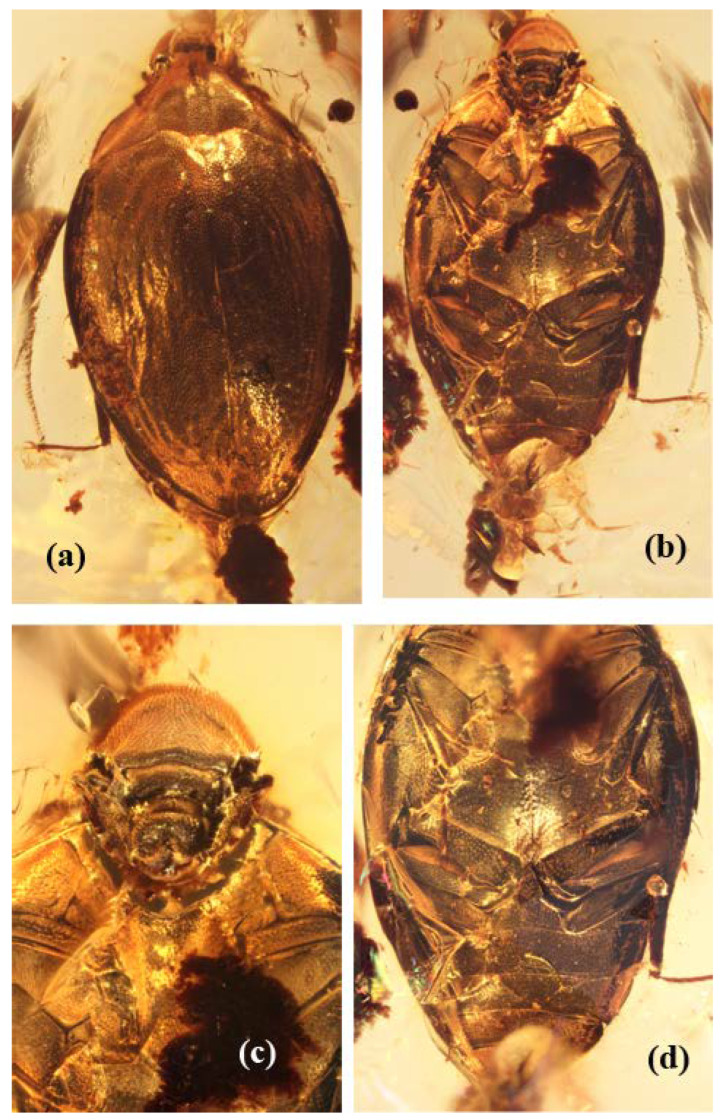
*Burmochares groehni* Kirejtshuk et Prokin, **gen. et sp. nov.**, holotype (GPIH no. 5071, CCGG no. 20037, LIBH): (**a**) habitus, dorsal view; (**b**) habitus, ventral view; (**c**) anterior part of body, ventral view; (**d**) posterior part of body, ventral view. Body length 2.85 mm.

**Figure 2 insects-13-00891-f002:**
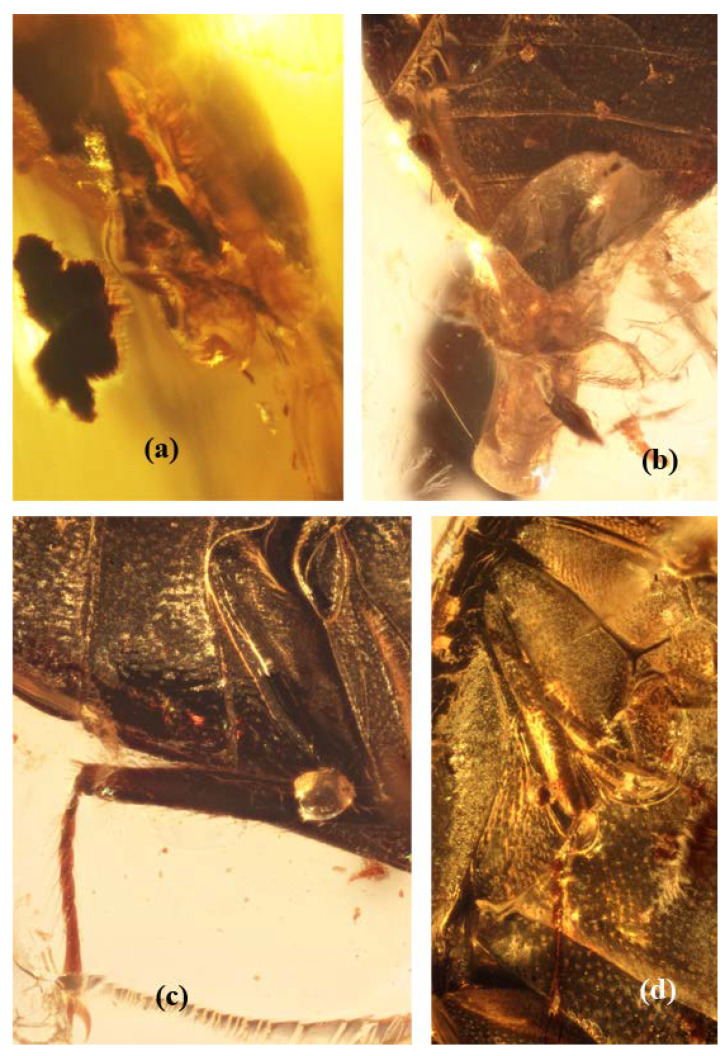
*Burmochares groehni* Kirejtshuk et Prokin, **gen. et sp. nov.**, holotype (GPIH no. 5071, CCGG no. 20037, LIBH): (**a**) abdominal apex, lateral view; (**b**) abdominal apex, ventral view; (**c**) hind leg; (**d**) mid leg. Body length 2.85 mm.

**Figure 3 insects-13-00891-f003:**
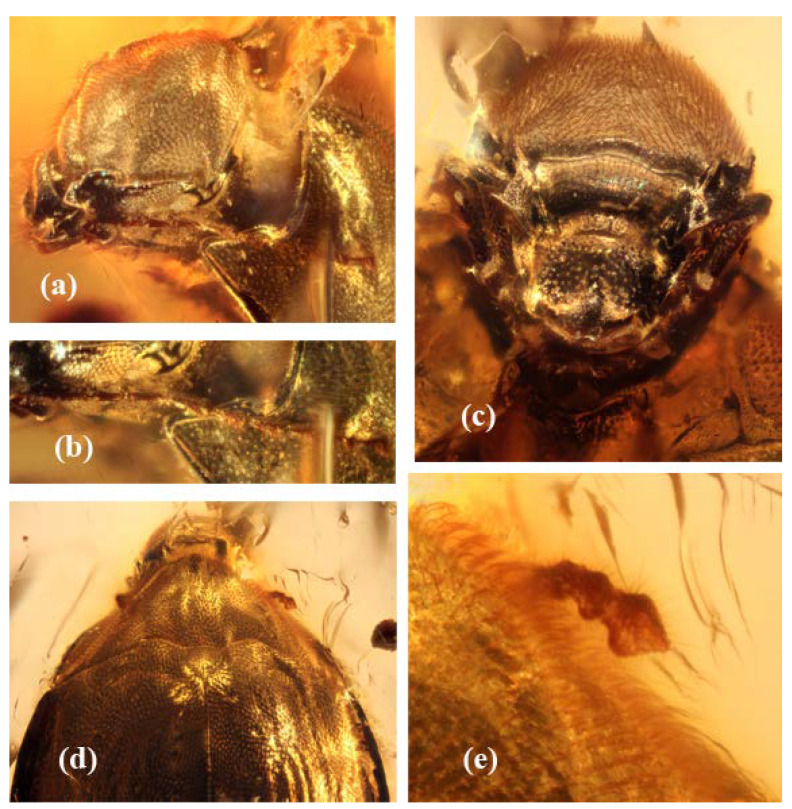
*Burmochares groehni* Kirejtshuk et Prokin, **gen. et sp. nov.**, holotype (GPIH no. 5071, CCGG no. 20037, LIBH): (**a**) head with right antenna, laterally; (**b**) right antenna; (**c**) head ventrally; (**d**) head, pronotum, apical segments of the left antenna, scutellum and the base of elytra, dorsally; (**e**) apical segments of the left antenna. Body length 2.85 mm.

## Data Availability

The new data were analyzed in this study and are openly available.

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
