# Peer review of "New Genus and Species of Limnichines from the Cretaceous Amber of Myanmar and Taxonomic Notes on the Family Limnichidae (Coleoptera, Polyphaga)†"

_insects, 2022, doi:10.3390/insects13100891_

Round 1

Reviewer 1 Report

Review of  “First fossil representative of the family Torridincolidae (Coleop-

tera, Myxophaga) from the Cretaceous amber of Myanmar with 3

notes on myxophagan fossil records” by Kirejtshuk & Prokin

This is an interesting paper describing a new genus and summarizing the fossil record of the Myxophaga.

Characters that define present day members of the Torridincolidae should be included in the Introduction

Reference to figures should be given in the text throughout the description and in other specific areas.

 A drawing clarifying the structure of the head (Fig. 1C) and the tarsal segments (Fig. 2C) would be good to include, especially since I count 6 tarsal segments.

The authors say that the integument is without sculptured elements, yet faint punctures on both dorsal and ventral surfaces and ridges on the dorsal surface are evident.

Regarding the sex, the authors should state whether they believe it to be a female, a male or that the sex is unknown.

The long discussion on the morphology and relationship of Haplochelus, Lepichelus, and Lepicerus is interesting but could be shortened.  The taxonomic conclusions summarize the author’s ideas well enough.

It is up to the editor whether the new “ruling” for the publication of Burmese fossils collected after 2017 is mentioned.  Since I do not believe in the new ruling, it does not matter to me.

Author Response

Dear colleague

Many thanks for your valuable review. The indications on figures in the text of the description were done and some new pictures were added. New data made it possible to re-interpret the fossil specimen examined and, therefore, the most part of general consideration included will be published in another paper. At this time we came to conclusion that the new genus and species described in this paper belong to a peculiar group of the Limnichinae which is already known after the previous studies of one modern genus and one genus from Baltic amber (additionally proposed by the authors of this paper). The explanations of our previous wrong interpretation were made in the new version of our paper.

Reviewer 2 Report

This is an interesting paper dealing with the description of the oldest known Torrindicolidae in the world. The authors present an adequate description and placement of the taxon within the family, with good photos. However, the text has some issues related to style, the lack of inclusion of figure citations (in fact, figures are never cited in the manuscript at all). The introduction is poor and need to be heavily improved, both in language and content. The discussion section could have a subtitle to indicate it is about the position of other myxophagans. My recommendation is that the manuscript needs some minor editing and it needs to be tighten up before it is ready for publication.

Author Response

Dear colleague

Many thanks for your valuable review. The indications on figures in the text of the description were done and some new pictures were added. New data made it possible to re-interpret the fossil specimen examined and, therefore, the most part of general consideration included will be published in another paper. At present we came to conclusion that the new genus and species described in this paper belong to a peculiar group of the Limnichinae which is already known after the previous studies of one modern genus and one genus from Baltic amber (additionally proposed by the authors of this paper). The explanations of our previous wrong interpretation were made in the new version of our paper. As to recommendation to increase of our Introduction, we regard that the part of the paper should be as laconic as possible and serve to explain purpose and main ideas included.

Reviewer 3 Report

The fossil is clearly misidentified in the familial level. The authors did not provide any single reason for assigning it to Myxophaga or Torridincolidae. The fossil it does not like anyhow "myxophagan" to me, and my first impression was it is more likely to be a limnichid (Myxophaga). The ventral morphology does not look like those torridincolids. One thing which seems to be of use - on their photo 1c, the ventral view of prothorax is seen very well - and the prosternum faces hypomeron directly, and there is no trace of propleuron (which normally in Myxophaga is present next to the posterior half of hypomeron). This can, in my opinion, exclude Myxophaga very clearly.  The Ms in the current form should be rejected.   

Author Response

The authors express many thanks for the reviewer’s time in the reviewing of our manuscript. Probably the note on “any single reason for assigning it to Myxophaga or Torridincolidae” is occasional as a sequence of the reviewer’s carelessness. The diagnostic characters of the suborder Myxophaga are rather ambiguous and contradictional. The suborder was erected as a compromise between some indications on relations of this group to Adephaga (mostly venation of hind wing) and some indications on relations of it to Polyphaga (mostly imaginal thoracic features and different larval features). Therefore, the most suborder diagnostic  characters are not visible in fossil specimens in principal. Moreover this contradiction between characters remains unsolved till now. Besides, one of argument for joining of the Myxophaga and Polyphaga is the complete invagination of their propleura inside the thorax, although one myxophagan and some polyphagans demonstrate a part of propleura externally as exclusions in the diagnostic syndromes of both suborders. The following reviewer sentence can produce a confusion: “there is no trace of propleuron (which normally in Myxophaga is present next to the posterior half of hypomeron).” By the by, the characters of the torridincollids are mentioned in the first and current versions of MS.

Nevertheless, this review was efficient enough because we asked the owner of this specimen to make new pictures and obtained some new very important arguments that our initial dilemma in the attribution of the specimen examined needs to be re-considered. At the moment we completely sure in our interpretation because the specimen examined is included in a peculiar group of the Limnichinae which is already known after the previous studies of one modern genus and one genus from Baltic amber (additionally proposed by the authors of this paper). The explanations of our previous wrong interpretation were made in the new version of our paper.

Round 2

Reviewer 3 Report

The revised ms has been greatly improved. The most significant changes lie in the classifications. In the revised version, the authors correctly placed the burmite fossil in Polyphaga, rather than Myxophaga. I still have two concerns about the ms though: 

1. A new genus Hernandochares gen. nov. is proposed for Platypelochares electricus Hernando, Szawaryn 21 et Ribera, 2018 from the Eocene Baltic amber. I disagree with such a taxonomic act, when the authors did not study the holotype but just based on published photos. And this is not the focus of the paper at all.  

2. The concept 'Dryopoidea' is adopted here, but the authors failed to provide a classification history of this superfamily. Dryopoidea has regarded as a part of Byrrhoidea in recent works, but recent phylogenomic studies suggested that Byrrhidae are not closely to members of 'Dryopoidea'. Please revise. 

Author Response

Thanks for the second review. Two notes with critics mentioned and require our comments:

Note 1: As to proposal new genus Hernandochares gen. nov. (reviewer: “this is not the focus of the paper at all)”, we suspect that the reviewer did not pay attention to our Discussion and our Comparision. There we explained the circumstances which forced us to propose a new taxon. We could agree with reviewer that it would be better not to propose new taxa without knowledge on composition of them (not only with generic rank). Probably the reviewer was recently shocked by one paper published this year with proposals of many family and ordinal taxa, which were not provided with real reasons, knowledge and grounding. We completely agree with the reviewer that this publication produced a lot of information noise, including problems and confusions in subjects where they are absent before. If so, we understand and share the reviewer’s opinion… However, in the case with Hernandochares gen. nov. the situation is alternative because we were lucky to have in our disposal rather professional descriptions with perfect illustrations of all modern and Eocene limnichid members related to the fossil species described in our paper. All characters taken for the diagnosis of Hernandochares gen. nov. shown in the illustrations of the papers cited in our paper.

Note 2: In our paper the principal base of dryopoids and byrrhoids -proposed and grounded by Hinton and Crowson is shortly considered. The same concerns the current grounding of classificational splits of the infraorder Elateriformia and particularly grounding of the superfamily Dryopoidea made by Crowson (it is also mentioned in our manuscript). After Crowson some professionals on this groups published some additional arguments for grounding the proposed dryopoid groups (they were mentioned with the references on limnichid taxa). Besides, after Crowson there were also researchers who made many attempts to revise these groups by computer sorting them in accordance with mostly subjective characters organized in the matrices with modifications of the Lawrence’s ones (it was also mentioned). The reviewer proposed “Please revise” the dryopoids. It would be a really quite great contribution but is far from the topic of this paper. To make this needs lot of intensive studies during many years. Nevertheless, thanks for reviewer’s proposal. We restricted our goal only by description of one new species and two new genera.